# Anthropometric and neurocognitive consequences of *Campylobacter*, enterotoxigenic *Escherichia coli*, and norovirus: A systematic review

Patricia B. Pavlinac[1,2]*, Gregory K. Zane[2], Ibrahim Khalil[1], Elizabeth T. Rogawski McQuade[3], James A. Platts-Mills[4], Mathias Lalika[1], Fatima H. Al-Shimari[1], Priyanka Shrestha[1], Birgitte K. Giersing[5], Mateusz Hasso-Agopsowicz[5]

1 Department of Global Health, University of Washington, Seattle, Washington, United States of America, 2 Department of Epidemiology, University of Washington, Seattle, Washington, United States of America, 3 Department of Epidemiology, Emory University, Atlanta, Georgia, United States of America, 4 Division of Infectious Diseases and International Health, University of Virginia, Charlottesville, Virginia, United States of America, 5 Department of Immunizations, Vaccines, and Biologicals, World Health Organization, Geneva, Switzerland

* ppav@uw.edu

## Abstract

### Objectives

Synthesizing the evidence of the longer-term consequences of enteric pathogens, such as stunted growth and suboptimal neurodevelopment, is a key step to articulating the value of, and generating demand for, vaccines.

### Methods

We conducted a systematic review of published literature documenting associations of three leading causes of diarrhea (enterotoxigenic *Escherichia coli* [ETEC], norovirus, and *Campylobacter* species [sp.]) with prospective anthropometric and neurocognitive outcomes in children under five years (PROSPERO CRD42024600676).

### Results

Thirty publications were included, including several reporting on data from the same underlying cohort; 16 publications included outcomes associated with *Campylobacter*, 12 ETEC, and 7 norovirus. There was large variation in how studies reported outcomes, exposure groups, and timeframes of association. There was modest evidence of linear growth detriments associated with all three pathogens, modest evidence of *Campylobacter* limiting weight gain, and no evidence of detrimental impacts of these pathogens on wasting or neurodevelopment, albeit these two outcomes were rarely reported.

**Data availability statement:** All relevant data are in the manuscript and its Supporting information files.

**Funding:** 1. This systematic review was funded by the Bill & Melinda Gates Foundation (INV-000518) granted to the World Health Organization. The Strategic Analysis, Research, and Training (START) Center at the University of Washington also supported this work. START is a collaborative effort with, and is supported by, the Bill and Melinda Gates Foundation (grant # OPP1155935). The funder of the study proposed the study design but had no role in data collection, analysis, or interpretation.

**Competing interests:** The authors have declared that no competing interests exist.

## Conclusion

Differences in outcome definitions, comparison groups, and timeframes prohibited meta-analysis and emphasize the need for more standardization of reporting anthropometric and neurocognitive outcomes following enteric pathogen infection. Randomized controlled trials of efficacious pathogen-specific interventions may help to address challenges with confounding and reverse causality in observational studies.

## Author summary

We reviewed previously conducted research studies to see if infections with common diarrhea-causing infections have long lasting effects on growth and brain development in young children. We specifically looked at published studies that tested for norovirus, *Campylobacter*, and enterotoxigenic *E.coli* in stool samples and followed children over time to evaluate changes in weight, height, and neurodevelopmental outcomes. Thirty studies met our inclusion criteria, some from the same study group, and studies varied widely in terms of how they assessed these associations and the duration of follow-up. There was modest evidence that each pathogen was associated with slower height growth and some evidence that *Campylobacter* was associated with restricted weight gain following infection. Although we did not find evidence that these infections led to measurable delays in neurocognitive development, these outcomes were rarely assessed, possibly because they require long follow-up to assess. We recommend standardized reporting of comparison groups and timeframes to enable synthesizing data on these complex relationships. Conducting pathogen-specific intervention trials, controlling for differences between those with and without the intervention, will clarify the extent to which these infections impact child growth and development, and help inform policy decisions around priorities for such interventions.

## Introduction

Diarrheal diseases cause over 400,000 deaths per year in children under 5 living in low and middle-income countries (LMICs) [1]. Through pathways involving local and systemic inflammation and intestinal destruction, many diarrhea-causing pathogens also contribute to longer- term morbidities such as wasting, stunting, reduced school performance, and reduced earning potential, consequences which are estimated to increase deaths attributed to diarrhea by 25% [2]. Also, more than half of diarrhea episodes among children in LMICs are treated with antibiotics making diarrhea a major contributor to antimicrobial resistance (AMR) [3].

Safe and effective rotavirus vaccines are available globally and rotavirus-attributed diarrhea burden and consequences are expected to continue declining as more countries adopt rotavirus vaccines into their routine EPI schedule. Vaccines

addressing other deleterious enteric pathogens are needed to further reduce diarrhea-attributed mortality, morbidity, and long-term consequences. Decisions to develop, introduce, and use vaccines balance on burden, both of acute disease and associations with long-term morbidity, feasibility of developing vaccines, as well as their expected uptake and use in countries.

An expert working group convened by the World Health Organization (WHO) identified four priority enteric pathogens based on morbidity, feasibility of vaccine development, contribution to AMR, and expected vaccine uptake: *Shigella*, enterotoxigenic *Escherichia coli* (ETEC), *Campylobacter jejuni,* and norovirus [4]. *Shigella*, St-ETEC, and *Campylobacter,* are bacteria prone to antibiotic resistance that are common causes of diarrhea, growth faltering, and antibiotic use. Norovirus, a viral pathogen, is an important cause of diarrhea and antibiotic use in both high-and low- to middle-income countries also with links to linear growth faltering.

Recent systematic reviews were conducted summarizing the attribution of specific enteric pathogens to diarrhea [5] as well as the case-fatality rates of several enteric pathogens [6]. We recently conducted a systematic review of the longer-term consequences of *Shigella* infection in young children, outlining evidence of this gram-negative bacterium's impact on persistent diarrhea, linear growth faltering, and potentially catastrophic health spending [7]. Here we report findings from a similar systematic review of studies evaluating the associations of ETEC, *Campylobacter*, and norovirus with growth and neurocognitive outcomes. As vaccines for these pathogens continue to move down the pipeline, this evidence summary can inform prioritization based on likelihood of averting not only diarrhea, but also growth faltering and other long-term morbidities.

## Methods

PubMed, Embase, LILACS, and SciELO were searched for articles published between January 1, 1980 to August 21, 2024 using terms outlined in S1 Table (Prospero: CRD42024600676). Using COVIDENCE (Veritas Health Innovation, Melbourne, Australia), titles and abstracts were reviewed by two independent reviewers using pre-specified inclusion criteria. These studies were further probed by a full text review performed by the two independent reviewers, to finalize the list of included studies. Publications had to meet all of the following criteria to be included: included children under 5 years with one or more priority pathogens (*Campylobacter*, ETEC, norovirus) identified from a fecal sample (irrespective of the presence of diarrhea); published from 1980 on; and included prospective (not cross-sectional) ascertainment of one or more outcomes of interest. Outcomes of interest included: anthropometric measurements including weight, height, length/heigh-for-age z-score [LAZ/HAZ], weight-for-age z-score [WAZ], weight-for-height z-score [WHZ], and mid-upper arm circumferences [MUAC]), stunting [LAZ/HAZ]<-2, wasting [WHZ<-2 in any age group or MUAC<12.5 in over 6 month olds]), and underweight (WAZ<-2) as well as neurodevelopmental outcomes.

The following study-level information was abstracted from included manuscripts: publication information, study design, study setting, study population, participant age-range, recruitment time frame, and pathogen detection method (bacterial culture, enzyme linked immunoassays [ELISA], and/or polymerase chain reactions [PCR]). Within a study, outcome measures and/or measures of association were abstracted for each of the following strata: time between pathogen detection and outcome ascertainment, unadjusted and adjusted (for all reported sets of confounders), by various exposure group categorization, including whether the pathogen was identified during diarrhea or sub-clinically, and by country.

We utilized a 10-point quality rating system adapted from the Strengthening the Reporting of Observational Studies in Epidemiology (STROBE) Statement [8]. Points were allocated based on key metrics reported in the included article and/or in referenced parent articles, such as relating to adequate description of the study population, study design, statistical power, bias addressed through participant selection and/or statistically, and describing how missing data and losses to follow-up were accounted for (S2 Table). A maximum score of 10 points indicated the highest quality. Scores of eight to ten were considered good quality, scores of five to seven were considered fair, and scores of less than five were considered poor quality.

## Results

We screened 3,784 non-duplicate titles and 95 full text articles, of which 30 met inclusion criteria. Several publications utilized data from the same parent study; eleven publications were from the MAL-ED study [9], six from the GEMS [10], two from the PROVIDE study [11], and two from the same surveillance study in Peru (**Table 1 and Fig 1**). Included studies were conducted as early as 1978–1979 [13] and as recently as 2020–2021 [20]. When considering unique cohorts, studies were most commonly conducted in Africa (n = 8) and Asia (n = 5), followed by South America, with Bangladesh having the most representation among countries (n = 7), followed by India (n = 2), Peru (n = 2), and Pakistan (n = 2). Prospective outcomes were assessed, most commonly, between 2 and 24 months, with some studies including 60-month follow-up and one with up to 96 months of follow-up. Linear growth (change in LAZ/HAZ; absolute change in length; risk of stunting) was the most assessed outcome category, followed by weight gain (change in WAZ; absolute change in weight; risk of underweight), and change in WHZ/wasting. Neurodevelopmental outcomes were only assessed in two studies. The STROBE adapted quality ratings of included study scores ranged from five to ten, with 20/30 studies meeting the good quality assessment. Four [15,25,32,37] out of the five highest scores [15,25,31,32,37] corresponded to the manuscripts reporting from large multi-country studies on multiple pathogens (GEMS and MAL-ED)—enabling within-study comparisons across pathogens.

### Campylobacter

Twenty-two publications reported on consequences of *Campylobacter* (**Table 2**), with 12 reporting weight gain outcomes, five WHZ outcomes, 20 linear growth, and two on neurodevelopmental outcomes. Four out of the 12 publications that addressed weight change as measured by absolute weight or WAZ found a statistically significant association between *Campylobacter* and weight. Each *Campylobacter* detection at routine monthly stool sample collection in the MAL-ED cohort was associated with a 0.63 (95%CI: -0.79, -0.43) lower WAZ [34] over a 24-month period. Another publication using MAL-ED data found a 0.22 lower WAZ over a 24-month period when comparing WAZ between children with *Campylobacter* detection in every monthly stool sample during predetermined time intervals to those with no *Campylobacter* infections [15]. Each incident episode of *Campylobacter* diarrhea was associated with a 55.3g (95%CI: -102.1, -8.4) lower average weight two months later, but % days with *Campylobacter* from the same study population was not (-2.9g [95%CI: -7.9, 2.2) [30]. Wasting was only associated with *Campylobacter* in one of the five studies that assessed this outcome, reporting an average loss of 0.16 SD in WHZ after 60 days of follow-up among children 24–59 months with asymptomatic *Campylobacter* infection [26].

   *Campylobacter* had some evidence of association with length change, both in the short term (three months) [35,37,38] and long term (24 months) [12,15,24,28,34,37], albeit inconsistently and with effect sizes ranging from -0.90 to 0.006. Other studies found no statistically significant relationship between *Campylobacter* and linear growth [16,20,41]. There did not appear to be a consistent association with length measures based on whether or not the *Campylobacter* was detected with diarrhea compared to without. *Campylobacter*-attributed diarrhea was inconsistently associated with LAZ two months later [16,26] and not at longer time intervals such as 12 [20,41] and 24 months [20]. However LAZ was modestly associated with *Campylobacter* diarrhea in MAL-ED three months later [37]. *Campylobacter* diarrhea was also not associated with neurodevelopmental outcomes in the two studies that evaluated this outcome [20,40].

### ETEC

Twenty publications reported on consequences of ETEC (**Table 2**), with 14 reporting weight gain outcomes, seven wasting outcomes, 16 linear growth, and two on neurodevelopmental outcomes. Short term (2 months) absolute weight gain [13] was associated with ETEC-diarrhea in one Bangladeshi study but not in a study among Peruvian children [30] nor in the GEMS multicenter study [17]. Each ETEC infection detected during routine monthly collection over a 24 month period

**Table 1. Description of publications included in this review, listed alphabetically.**

| Study | Country | Recruitment Years | Study name (if applicable) | Age range at enrollment | Population description | Microbiologic confirmation method | Length of follow-up (months) | Total Sample Size | Pathogens of interest |
|---|---|---|---|---|---|---|---|---|---|
| Amour 2016 [12] | Bangladesh, India, Nepal, South Africa, Tanzania, Brazil, Peru | 2009-2012 | MAL-ED | 0-17 days | Healthy infants enrolled within 17 days of birth | ELISA | 24 | 1426 | *Campylobacter* |
| Black 1984 [13] | Bangladesh | 1978-1979 | Not named | 6-48 months | Bangladeshi children living in a village-based surveillance field research area | Culture | Not specified (at least 2 months) | 157 | ETEC |
| Bray 2019 [14] | Bangladesh | 2007-2010 | GEMS | 0-59 months | Cases: Children aged 0–59 months presenting to health facilities with moderate-to-severe diarrhea (MSD) Controls: age-, gender-, and community-matched from a demographic surveillance system within 14 days of case presentation. | Not specified but presumed to be qPCR | 2 | 3859 (cases and controls) | Norovirus |
| Caulfield 2017 [15] | Bangladesh, India, Nepal, South Africa, Tanzania, Brazil, Peru | 2009-2014 | MAL-ED | 0-17 days | Healthy infants enrolled within 17 days of birth | ELISA & PCR | 24 | 1291 | *Campylobacter* & ETEC |
| Das 2021 [16] | Bangladesh | 2007-2011 | GEMS | 0-59 months | Cases: Children aged 0–59 months presenting to Kumudini Hospital with moderate-to-severe diarrhea (MSD) during the 36-month period of the study. Controls: age-, gender-, and community-matched from a demographic surveillance system within 14 days of case presentation. | Culture | 2 | 1333 | *Campylobacter* |
| Das 2022 [17] | Bangladesh, India, Pakistan, The Gambia, Mali, Mozambique, Kenya | 2007-2011 | GEMS | 0-59 months | Cases: Children aged 0–59 months from the DSS catchment area presenting at health facilities within 7 days of the onset of a new and acute episode of moderate-to-severe diarrhea (MSD) Controls: age-, sex-, and community-matched children without diarrhea randomly selected from the DSS database within 14 days of each enrolled case | Multiplex qPCR | 2 | 1539 | ETEC |

*(Continued)*

**Table 1.** (Continued)

| Study | Country | Recruit-ment Years | Study name (if applicable) | Age range at enroll-ment | Population description | Microbio-logic con-firmation method | Length of follow-up (months) | Total Sample Size | Patho-gens of interest |
|---|---|---|---|---|---|---|---|---|---|
| Das 2024 [18] | Bangla-desh, India, Pakistan, The Gambia, Mali, Mozambique, Kenya | 2007-2011 | GEMS | 0-59 months | Cases: Children aged 0–59 months from the DSS catchment area presenting at health facil-ities within 7 days of the onset of a new and acute episode of moderate-to-severe diarrhea (MSD) Controls: age-, sex-, and community-matched children without diarrhea in the previous 7 days, randomly selected from the DSS database | Multiplex RT-PCR | 2 (50–90 days) | 5572 | Norovirus |
| Diaz 2023 [19] | Haiti | 2020-2021 | Not named | 6–36 months | Cases: defined using a standard epidemiological definition (care-giver report of three or more liquid/semi-liquid stools in a 24-hour period over the preceding three days). Control: those without diarrhea at enrollment. Diarrheal symptoms were assessed again, one month later, and individu-als were further sorted by follow-up status. | Qualitative PCR | 1 | 195 (136 completed follow-up) | ETEC |
| Donowitz 2021 [20] | Bangladesh | 2014-2016 | Not named | 0-2 years | Children enrolled in a birth cohort (enrolled within first 7 days of life) | qPCR | 24 | 250 | *Campy-lobacter,* ETEC, norovirus |
| George 2017 [21] | Bangladesh | 2014 | Not named | 6-30 months | Children living in the icddr,b DSS aged 6–30 months | qPCR | 9 | 203 | *Campy-lobacter,* ETEC |
| George 2023 [22] | Democratic Republic of the Congo | 2018-2019 | REDUCE | <5 years | Cases: defined using a standard epidemiological definition (care-giver report of three or more liquid/semi-liquid stools in a 24-hour period over the preceding three days). Control: those without diarrhea at enrollment. Diarrheal symptoms were assessed again, one month later, and individu-als were further sorted by follow-up status. | qPCR | 6 | 236 | *Campy-lobacter,* ETEC |

*(Continued)*

**Table 1.** (Continued)

| Study | Country | Recruit-ment Years | Study name (if applicable) | Age range at enroll-ment | Population description | Microbio-logic con-firmation method | Length of follow-up (months) | Total Sample Size | Patho-gens of interest |
|---|---|---|---|---|---|---|---|---|---|
| González-Fernández 2023 [23] | Pakistan | 2010-2012 | MAL-ED | ≤ 17 days | Healthy infants enrolled within 17 days of birth | Culture, ELISA, PCR | < 66 | 277 | *Campy-lobacter*, ETEC, norovirus |
| Haque 2019 [24] | Bangladesh, India, Nepal, South Africa, Tanzania, | 2009-2012 | MAL-ED | ≤ 17 days | Healthy infants enrolled within 17 days of birth | qPCR & ELISA | 24 | 1715 | *Campylo-bacter* |
| Haque 2023 [25] | Bangladesh, Brazil, India, Nepal, Peru, Pakistan, South Africa, Tanzania | 2009-2012 | MAL-ED | ≤ 17 days | Healthy infants enrolled within 17 days of birth | qPCR (TaqMan) | 24 | 1,715 | ETEC |
| Hossain 2023 [26] | Bangla-desh, India, Pakistan | 2007-2011 | GEMS | 0-59 months | Cases: Children aged 0–59 months from the DSS catchment area presenting at health facilities within 7 days of the onset of a new and acute episode of moderate-to-severe diarrhea (MSD) Controls: age-, sex-, and community-matched children | Culture | 2 | 22,567 (1843 positive for *Campylo-bacter*) | *Campylo-bacter* |
| Iqbal 2019 [27] | Pakistan | 2012-2015 | Not named | 0-14 days | Children enrolled in a birth cohort (enrolled within first 14 days of life) | qPCR | 18 | 272 | *Campy-lobacter*, ETEC, norovirus |
| Kabir 2022 [28] | Pakistan | 2016-2018 | SEEM | 0-14 days | Healthy newborns enrolled within 14 days of birth | qPCR | 24 | 416 | *Campy-lobacter*, ETEC |
| Lee, 2013 [29] | Peru | 2002-2006 | Not named but same cohort as Lee 2014 | 0-72 months | Community-based children living in the cen-sused population | Culture | Variable (recruit-ment ongo-ing and children aged out at 72 months) | 433 | *Campylo-bacter* |
| Lee, 2014 [30] | Peru | 2002-2006 | Not named but same cohort as Lee 2013 | 0-72 months | Community-based children living in the cen-sused population | Culture & PCR | Variable (recruit-ment ongo-ing and children aged out at 72 months) | 433 | *Campy-lobacter*, ETEC |
| Luoma 2023 [31] | Malawi | 2009-2011 | iLiNS-DYAD-M | Pregnant women enrolled, and child entered study at birth | Children enrolled in study with data available at 18 months (infection evalua-tion) and 24 months (LAZ) | RT-PCR | 6 | 604 | *Campy-lobacter*, norovirus |

*(Continued)*

**Table 1.** (Continued)

| Study | Country | Recruitment Years | Study name (if applicable) | Age range at enrollment | Population description | Microbiologic confirmation method | Length of follow-up (months) | Total Sample Size | Pathogens of interest |
|---|---|---|---|---|---|---|---|---|---|
| Nasrin 2021 [32] | The Gambia, Mali, Mozambique, Kenya, Pakistan, Bangladesh, India | 2007-2011 | GEMS | 0-59 months | Children presenting to health facilities with acute diarrhea and one or more of: dehydration, dysentery, or indication for hospitalization (MSD) | Culture & PCR | 2 (range 1.5-3) | 7545 | ETEC |
| Pajuelo 2024 [33] | Peru | 2016-2019 | Not named | 0-35 days | Healthy infants enrolled within 35 days of birth | qPCR | 24 | 345 | ETEC |
| Palit 2022 [34] | Bangladesh, India, Nepal, South Africa, Tanzania, Brazil, Peru | 2009-2012 | MAL-ED | 0-17 days | Healthy infants enrolled within 17 days of birth | qPCR | 24 | 1715 | *Campylobacter*, ETEC, norovirus |
| Platts-Mills 2014 [35] | Tanzania | 2009-2012 | MAL-ED | 0-17 days | Healthy infants enrolled within 17 days of birth | qPCR | 24 | 61 | *Campylobacter,* ETEC |
| Platts-Mills 2017 [36] | Bangladesh | 2009-2012 | PROVIDE | 6-23 months | Case: Children aged 6–23 mo presenting to a community malnutrition clinic with a weight-for-age z score (WAZ) <−2 Control: frequency matched to cases by age, sex, and area of residence, and a WAZ >−1 same neighborhood | qPCR | 60 | 928 | *Campylobacter,* ETEC, norovirus |
| Rogawski, 2018 [37] | Bangladesh, India, Nepal, South Africa, Tanzania, Brazil, Peru | 2009-2012 | MAL-ED | 0-17 days | Healthy infants enrolled within 17 days of birth | qPCR | 24 (59 months in subset) | 1469 | *Campylobacter,* ETEC, norovirus |
| Rouhani 2020 [38] | Peru | 2009-2012 | MAL-ED | 0-17 days | Healthy infants enrolled within 17 days of birth | Culture and ELISA | 24 (59 months in subset) | 271 | *Campylobacter* |
| Sanchez, 2020 [39] | Bangladesh | 2010-2012 | MAL-ED | 0-17 days | Healthy infants enrolled within 17 days of birth | Culture and ELISA | 24 | 265 | *Campylobacter* |
| Scharf 2023 [40] | Brazil, Tanzania, South Africa | 2009-2012 | MAL-ED | 0-17 days | Healthy infants enrolled within 17 days of birth | qPCR | 72-96 | 451 | *Campylobacter,* ETEC, norovirus |
| Schnee, 2018 [41] | Bangladesh | 2011-2014 | PROVIDE | 0-7 days | Healthy infants enrolled within 7 days of birth | qPCR | 24 | 603 | *Campylobacter*, ETEC, norovirus |

in one MAL-ED publication was associated with an average 0.65 lower WAZ (95 CI: -0.78, -0.42, p=0.02) [34] whereas another study in the same cohort found no consistent relationship focusing specifically on LT-ETEC [15]. Wasting did not appear to be associated with ETEC over a nine-month [21] and 24-month period [37] but site specific analyses of the MAL-ED study found an increased likelihood of wasting associated with ETEC in Tanzania [25].

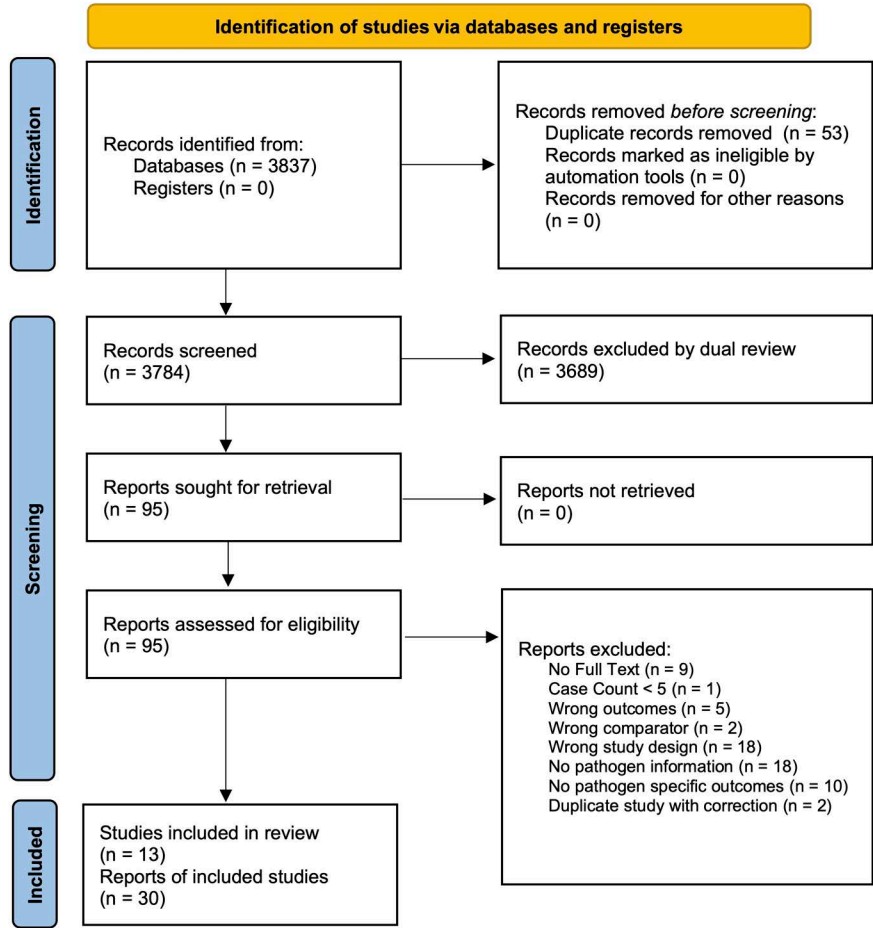

**Fig 1. Study inclusion PRISMA flow diagram [14].**

ETEC was significantly associated with linear growth in three of 17 studies evaluating this association. Children aged 12–23 months (but not 0–11 month olds) with St-ETEC-attributable moderate-to-severe diarrhea (MSD) had a 0.12 greater loss in LAZ in the 60-days following their diarrhea episode than children with MSD not attributable to St-ETEC (95%CI: -0.17, -0.06), (p < 0.001) [32]. Cumulative asymptomatic ETEC was associated with linear growth three months after in a single site of the MAL-ED study [35] but not across all MAL-ED sites over the 24 and 60-month follow-up period (-0.04 [95%CI: -0.07, -0.01] [37]. ETEC-attributable diarrhea was inconsistently associated with linear growth with a statistically significant association between ETEC-attributable diarrhea and linear growth three months after [37] in MAL-ED and in a Peruvian cohort over nine months [30] but not in two other studies looking at LAZ after 12 months [20,41]. ETEC was not associated with neurodevelopmental outcomes in the studies evaluating this outcome [20,40].

## Norovirus

**Table 2** describes the eight studies evaluating norovirus' associations with long-term outcomes (weight gain [n = 5], wasting [n = 3], length [n = 9], and neurocognitive measures [n = 2]). In the five studies that evaluated norovirus' association with weight gain, one from the MAL-ED cohort demonstrated evidence of a statistically significant association between norovirus GII infection detected during routine monthly stool collection in 24 month change in WAZ (-0.39 [95%CI: -0.49, -0.28],

**Table 2. Anthropometric and neurocognitive outcomes among children with *Campylobacter*, ETEC, or norovirus infections.**

| Outcome | Exposure/out-come interval | Paper | Study (if applicable) | Comparison groups | Effect measure |
|---|---|---|---|---|---|
| **a. *Campylobacter*** | | | | | |
| **Weight Gain Outcomes** | | | | | |
| **Change/difference in WAZ** | | | | | **Mean △ in WAZ(95% CI)** |
| | 2 months | Das 2021 [16] | GEMS | Comparing moderate-to-severe diarrhea cases and controls with *Campylobacter* detected to those without *Campylobacter* detected | 0.05 (-0.04, 0.14); p = 0.257 |
| | | Hossain 2023 [26] | GEMS | Change in WAZ among those with symptomatic *Campylobacter* infection aged 0–11 months, 12–23 months, and 24–59 months | 0-11mo: -0.12 (-0.27, 0.03), p = 0.114; 12–23mo: -0.03 (-0.14, 0.08), p = 0.580; 24–59mo: -0.03 (-0.16, 0.09) p = 0.580 |
| | | [26] | | Change in WAZ among those with asymptomatic *Campylobacter* infection aged 0–11 months, 12–23 months, and 24–59 months | 0-11mo: 0.06 (-0.05, 0.16), p = 0.302; 12–23mo: -0.002 (-0.11, 0.10), p = 0.969; 24–59mo: -0.15 (-0.24, -0.05), p = 0.002 |
| | 5 months | Platts-Mills 2017 [36] | PROVIDE | Comparing children with and without *Campylobacter* detected | Not reported (see Fig 3#); P-value reported to be not significant (but exact value not reported); p-value NR |
| | 6 months | George 2023 [22] | REDUCE | Comparing children with and without *Campylobacter* detected, by 1.) presence vs. absence OR 2.) log-transformed presence vs. absence | 1.) 0.16 (−0.05 to 0.37) 2.) 0.01 (−0.01 to 0.02) |
| | 12 months | Kabir 2022 [28] | SEEM | Change in WAZ at 12 months among those 1.) with versus 2.) without *Campylobacter* detected at 3–6 months | 1.) -0.55 (IQR: -1.3, 0.30) vs. 2.) -0.35 (IQR: -1.14, 0.59), p = 0.200 |
| | | [28] | | Change in WAZ at 12 months among those 1.) with versus 2.) without *Campylobacter* detected at 9 months | 1.) -0.61 (IQR: -1.34, 0.29) vs. 2.) -0.12 (IQR: -0.87, 0.83), p = 0.002 |
| | 24 months | Caulfield 2017 [15] | MAL-ED | Comparing children with *Campylobacter* detected in every period compared to children with no *Campylobacter* exposure (with periods defined as enrollment to 2 months, 3–5 months, 6–8, 9-11m, 12-17m, and 18-24m) | -0.22 (or 0.22 ± 0.15 kg lighter); p-value = 0.14 |
| | | Palit 2022 [34] | MAL-ED | Per each *Campylobacter* infection detected during routine monthly stool collection | -0.63 (-0.79, -0.43), p < 0.001 |
| | | Rogawski 2018 [37] | MAL-ED | Comparing high (90th percentile) and low (10th percentile) *Campylobacter* prevalence in non-diarrheal stools | Not reported (see Figure S7#); confident interval crosses zero (p-value not significant) |
| | | Kabir 2022 [28] | SEEM | Change in WAZ at 24 months among those 1.) with versus 2.) without *Campylobacter* detected at 3–6 months | 1.) -0.47 (IQR: -1.12, 0.46) vs. 2.) -0.23 (IQR: -1.04, 0.55), p = 0.370 |
| | | [28] | | Change in WAZ at 24 months among those 1.) with versus 2.) without *Campylobacter* detected at 9 months | 1.) -0.50 (IQR: -1.15, 0.35) vs. 2.) -0.10 (IQR: -0.88, 0.62), p = 0.013 |
| **Mean Change in Weight (g)** | | | | | |
| | 2 months | Lee 2014 [30] | | Per incident episodes of *Campylobacter* diarrhea | -55.3 (95%CI: -102.1, -8.4); p = 0.021 |
| | | | | Per % of days spent with *Campylobacter* diarrhea | -2.9 (95%CI: -7.9, 2.2); p = 0.267 |
| | 3 months | Lee 2013 [29] | | Per each additional symptomatic *Campylobacter* diarrhea episode (1 vs. 0 or 2+ vs. 1) | -43.9 (95% CI: -87.6, -0.1); p = 0.049 |

*(Continued)*

**Table 2.** (Continued)

| Outcome | Exposure/out-come interval | Paper | Study (if applicable) | Comparison groups | Effect measure |
|---|---|---|---|---|---|
| | | | | comparing children with asymptomatic *Campylobacter* to children with no *Campylobacter* detected | -67.7 (95% CI: -130.1, -5.2); p<0.05 |
| **Risk of Underweight (WAZ<-2)** | | | | | **Odds Ratio (95% CI)** |
| | 9 months | George 2018 [21] | | comparing likelihood of stunting during follow-up among those with *Campylobacter* at baseline to those without, after adjusting for age, age, caregiver educational level, breastfeeding, and family size | 1.18 (0.53, 2.61); p>=0.05 |
| | 54–66 months | González-Fernández 2023 [23] | MAL-ED | Univariate analysis comparing relative risk of underweight at follow-up between those with versus without *Campylobacter* at least once between 0–5 months of age, adjusted for gender, first weight and income | Relative Risk: 1.63 (1.09, 2.44), p=0.018 |
| | | [23] | | Multivariate analysis comparing relative risk of underweight at follow-up between those versus without *Campylobacter* at least once between 0–5 months of age, adjusted for gender, first weight, income, feeding practices, among others | Relative Risk: 1.78 (1.17, 2.70), p=0.007 |
| **Wasting Outcomes** | | | | | |
| **Mean Change/difference in WHZ** | | | | | **Mean △ in WHZ (95% CI)** |
| | 2 months | Das 2021 [16] | GEMS | Comparing moderate-to-severe diarrhea cases and controls with *Campylobacter* detected to those without Campylobacter detected | WHZ: 0.04 (-0.05, 0.12); p-value: 0.397 |
| | | Hossain 2023 [26] | GEMS | Change in WHZ among those with symptomatic *Campylobacter* infection aged 0–11 months, 12–23 months, and 24–59 months | 0-11mo: -0.03 (-0.18, 0.13), p=0.709; 12–23mo: -0.06(-0.18, 0.06), p=0.340; 24–59mo: 0.07 (-0.06, 0.19), p=0.294 |
| | | [26] | | Change in WHZ among those with asymptomatic *Campylobacter* infection aged 0–11 months, 12–23 months, and 24–59 months | 0-11mo: 0.10 (-0.003, 0.20), p=0.058; 12–23mo: -0.11 (-0.22, 0.003), p=0.058; 24–59mo: -0.16 (-0.26, -0.06), p=0.001 |
| | 6 months | George 2023 [22] | REDUCE | Comparing children with and without *Campylobacter* detected, by 1.) presence vs. absence OR 2.) log-transformed presence vs. absence | 1.) 0.11 (–0.17 to 0.40) 2.) 0.002 (–0.02 to 0.02) |
| | 24 moths | Rogawski 2018 [37] | MAL-ED | Comparing high (90th percentile) and low (10th percentile) in non-diarrheal samples | Not reported (See Figure S7#), confident interval crosses zero (p-value not significant) |
| **Risk of Wasting (WHZ<-2)** | | | | | **Odds Ratio (95% CI)** |
| | 9 months | George 2018 [21] | | Comparing likelihood of stunting during follow-up among those with *Campylobacter* at baseline to those without, after adjusting for age, age, caregiver educational level, breastfeeding, and family size | 0.64 (0.20, 2.09); p>=0.05 |
| **Linear Growth Outcomes** | | | | | |
| **Mean Change/difference in HAZ/LAZ** | | | | | **Mean △ in HAZ/LAZ (95% CI)** |
| | 2 months | Das 2021 [16] | GEMS | Comparing moderate-to-severe diarrhea cases and controls with *Campylobacter* detected to those without *Campylobacter* detected | HAZ: 0.02 (-0.07, 0.10); p-value=0.732 |
| | | Hossain 2023 [26] | GEMS | Change in HAZ among those with symptomatic *Campylobacter* infection aged 0–11 months, 12–23 months, and 24–59 months | 0-11mo: -0.19 (-0.36, -0.03), p=0.018; 12–23mo: -0.02 (-0.12, 0.08), p=0.696; 24–59mo: -0.16 (-0.28, -0.04), p=0.010 |

*(Continued)*

| Outcome | Exposure/out-come interval | Paper | Study (if applicable) | Comparison groups | Effect measure |
|---|---|---|---|---|---|
| | | [26] | | Change in HAZ among those with asymptomatic *Campylobacter* infection aged 0–11 months, 12–23 months, and 24–59 months | 0-11mo:-0.04 (-0.15, 0.07), p=0.515; 12–23mo: 0.04 (-0.06, 0.14), p=0.395; 24–59mo: -0.07 (-0.16, 0.02), p=0.136 |
| | 3 months | Platts-Mills 2014 [35] | MAL-ED | Per log10 higher quantity of *Campylobacter* infection in asymptomatic stool | -0.11; p-value=0.003 |
| | | Rogawski 2018 [37] | MAL-ED | Within child comparison between baseline and 3 months following *Campylobacter*-attributable diarrhea | -0.02; p-value <0.01 |
| | | Rouhani 2020 [38] | MAL-ED | Per 3 month interval (from birth to 9 months) associated with an increase of 10% in the proportion of surveillance stools with *Campylobacter* detected | -0.02; p-value <0.01 |
| | 6 months | Sanchez 2020 [39] | MAL-ED | Comparing children with and without asymptomatic *Campylobacter* infections identified up until 3 months of life | 0.006 (CI NR); p-value 0.853 |
| | | George 2023 [22] | REDUCE | Comparing children with and without Campylobacter detected, by 1.) presence vs. absence OR 2.) log-transformed presence vs. absence | 1.) 0.09 (–0.15 to 0.34) 2.) 0.004 (–0.01 to 0.02) |
| | | Luoma 2023 [31] | iLiNS-DYAD-M | Difference in LAZ at 24 months comparing children with positive vs negative test result for *Campylobacter* at 18 months | 0.17 (−0.01 to 0.36), p=0.07 (positive: −1.82±1.02 vs. negative: −1.65±1.09) |
| | 9 months | Sanchez 2020 [39] | MAL-ED | Comparing children with and without asymptomatic *Campylobacter* infections identified up until 6 months of life | 0.006 (CI NR); p-value 0.853 |
| | 12 months | Donowitz 2021 [20] | | Per 1 episode of diarrhea attributed to *Campylobacter* | -0.28 (-0.03, 0.58); p=0.08 |
| | | Sanchez 2020 [39] | MAL-ED | Comparing children with and without asymptomatic *Campylobacter* infections identified up until 9 months of life | -0.115; p<0.0005 |
| | | Schnee 2018 [41] | PROVIDE | Per *Campylobacter*-attributable diarrhea episode | −0.16 (95% CI: −0.32, −0.01); p-value NR |
| | | | | per *Campylobacter*-attributable diarrhea episode in the first 6 months of life | −0.51 (95% CI: −0.92, −0.10); p-value NR |
| | | Kabir 2022 [28] | SEEM | Change in HAZ at 12 months among those 1.) with versus 2.) without *Campylobacter* detected at 3–6 months | 1.) −0.64 (IQR: −1.52, 0.07) vs. 2.) −0.44 (IQR: −1.27, 0.17), p=0.160 |
| | | | | Change in HAZ at 12 months among those 1.) with versus 2.) without *Campylobacter* detected at 9 months | 1.) −0.67 (IQR: −1.57, −0.04) vs. 2.) −0.23 (IQR: −1.15, 0.36), p=0.005 |
| | 15 months | Sanchez 2020 [39] | MAL-ED | Comparing children with and without asymptomatic *Campylobacter* infections identified up until 12 months of life | -0.115 (CI NR); p<0.0005 |
| | 18 months | Iqbal 2019 [27] | | Comparing children with and without *Campylobacter* infection at 6 months of age | -0.36; SE: 0.15; p-value=0.014 |
| | | | | Comparing children with and without *Campylobacter* infection at 9 months of age | -0.10; SE: 0.14; p-value 0.455 |
| | | Sanchez 2020 [39] | MAL-ED | Comparing children with and without asymptomatic *Campylobacter* infections identified up until 15 months of life | 0.005 (CI NR); p-value 0.872 |
| | 24 months | Amour 2016 [12] | MAL-ED | Model predicted 24 month LAZ Score for high (90th percentile) and low (10th percentile) *Campylobacter* burden presented separately (not formally tested but confidence intervals do not overlap) | High: −1.82 (95% CI: −1.94, −1.70) Low: −1.49 (95% CI: −1.60, −1.38) |

*(Continued)*

**Table 2.** (Continued)

| Outcome | Exposure/out-come interval | Paper | Study (if applicable) | Comparison groups | Effect measure |
|---|---|---|---|---|---|
| | | Caulfield 2017 [15] | MAL-ED | Comparing children with *Campylobacter* detected in every period compared to children with no *Campylobacter* exposure (with periods defined as enrollment to 2 months, 3–5 months, 6–8, 9-11m, 12-17m, and 18-24m) | -0.27 (or 0.83±0.33 cm shorter), p=0.01 |
| | | Donowitz 2021 [20] | | Per 1 episode of diarrhea attributed to *Campylobacter* | -0.03 (95%CI: -0.23, 0.17); p=0.76 |
| | | Haque 2019 [24] | MAL-ED | Comparing 90th percentile *Campylobacter* burden vs. 10th percentile *Campylobacter* burden (burden defined as proportion surveillance stool samples with *Campylobacter* detected by EIA) | −0.31 (95% CI: −0.46, −0.15); p<0.001 |
| | | | | Comparing 90th percentile *Campylobacter* burden vs. 10th percentile *Campylobacter* burden (burden defined as proportion surveillance stool samples with Campylobacter detected by EIA) | −0.18 (95% CI: −0.30, −0.06); p=0.004 |
| | | | | Comparing 90th percentile *Campylobacter* burden vs. 10th percentile *Campylobacter* burden | Difference NR but are presented in Fig 4A# and Fig 4B# High: −1.82 (95% CI: −1.94, −1.70) low: −1.49 (95% CI: −1.60, −1.38) |
| | | Palit 2022 [34] | MAL-ED | Per each *Campylobacter* infection detected during routine monthly stool collection | -0.30 (-0.38, -0.21), p<0.001 |
| | | Rogawski 2018 [37] | MAL-ED | Comparing high (90th percentile) and low (10th percentile) *Campylobacter* prevalence in non-diarrheal samples | -0.17 (95%CI: -0.32, -0.01) |
| | | | | Comparing high (90th percentile) and low (10th percentile) *Campylobacter*-attributable diarrhea burden | Not reported (See Figure S4#), confident interval crosses zero (p-value not significant) |
| | | | | Per one log increase in *Campylobacter* quantity in non-diarrhoeal stools | -0.05 (95%CI: -0.09, 0.01) |
| | | Sanchez 2020 [39] | MAL-ED | Comparing children with and without asymptomatic *Campylobacter* infections identified up until 21 months of life | 0.006 (CI NR); p=0.872 |
| | | Schnee 2018 [41] | PROVIDE | Per *Campylobacter*-attributable episode | −0.15 (95% CI: -0.32, 0.02); p-value NR |
| | | Kabir 2022 [28] | SEEM | Change in HAZ at 24 months among those 1.) with versus 2.) without *Campylobacter* detected at 3–6 months | 1.) −0.85 (IQR: −1.53, 0.05) vs. 2.) −0.51 (IQR: −1.30, 0.04), p=0.140 |
| | | Kabir 2022 [28] | SEEM | Change in HAZ at 24 months among those 1.) with versus 2.) without *Campylobacter* detected at 9 months | 1.) −0.90 (IQR: −1.61, −0.08) vs. 2.) −0.38 (IQR: −1.02, 0.19), p-0.001 |
| | 60 months | Rogawski 2018 [37] | MAL-ED | Comparing high (90th percentile) and low (10th percentile) *Campylobacter* prevalence in non-diarrheal samples | Not reported (see Fig 4A#); confident interval crosses zero (p-value not significant) |
| | | | | Per one log increase in the mean quantity of *Campylobacter* per gram of stool | Not reported (See Fig 4B#); confident interval crosses zero (p-value not significant) |
| **Mean Change in Linear Growth (cm)** | | | | | |
| | 9 months | Lee 2014 [30] | | Per % of days spent with *Campylobacter* diarrhea | -0.029 (95%CI: -0.058, 0.001), p=0.052 |
| | | | | Per incident episode of *Campylobacter* diarrhea | -0.067 (95%CI: -0.127, -0.007) |

*(Continued)*

**Table 2.** (Continued)

| Outcome | Exposure/out-come interval | Paper | Study (if applicable) | Comparison groups | Effect measure |
|---|---|---|---|---|---|
| | 9 months | Lee 2013 [29] | | Per each additional symptomatic *Campylobacter* diarrhea episode (1 vs. 0 or 2+vs. 1) | -0.059 (95% CI: -0.12, 0.01), p=0.59 |
| | | | | Comparing children with asymptomatic *Campylobacter* to children with no *Campylobacter* detected | -0.01 (95% CI: -0.09, 0.07), p>=0.1 |
| **Risk of Stunting (HAZ<-2)** | | | | | **Odds Ratio (95% CI)** |
| | 9 months | George 2018 [21] | | Comparing likelihood of stunting during follow-up among those with *Campylobacter* at baseline to those without, after adjusting for age, age, caregiver educational level, breastfeeding, and family size | 0.92 (0.44, 1.90), p>=0.05 |
| | 54–66 months | González-Fernández 2023 [23] | MAL-ED | Univariate analysis comparing relative risk of stunting at follow-up between those with versus without *Campylobacter* at least once between 0–5 months of age, adjusted for gender, first weight and income | Relative Risk: 0.27 (0.91, 1.78), p=0.153 |
| **Neurodevelopmental Outcomes** | | | | | |
| **Change in Bayley-III Score** | | | | | |
| | 12 months | Donowitz 2021 [20] | | Per 1 episode of diarrhea attributed to *Campylobacter* | Cognition: 0.46 (-0.86 to 1.78), p=0.5 Language: -0.24 (-1.89 to 1.40), p=0.77 Motor abilities: -0.79 (-2.32 to 0.74), p=0.31 |
| **Change in Semantic Fluency** | | | | | |
| | 72-96 months | Scharf 2023 [40] | MAL-ED | Comparison of semantic fluency (Words/animals in a minute from NEPSY) between children with 1 or more episodes of diarrhea attributable to *Campylobacter* and children with 0 episodes attributable to *Campylobacter* between 1–24 months of age | Estimate: -0.60 (-1.38, 0.17), p=0.13 |
| **Change in Phonemic Fluency** | | | | | |
| | 72-96 months | Scharf 2023 [40] | MAL-ED | Comparison of phonemic fluency (Words beginning with S & F in a minute from NEPSY) between children with 1 or more episodes of diarrhea attributable to *Campylobacter* and children with 0 episodes attributable to *Campylobacter* between 1–24 months of age | Estimate: -0.23 (-0.97, 0.52), p=0.55 |
| **Change in Reasoning Skills** | | | | | |
| | 72-96 months | Scharf 2023 [40] | MAL-ED | Comparison of reasoning skills (Raven colored progressive matrices) between children with 1 or more episodes of diarrhea attributable to *Campylobacter* and children with 0 episodes attributable to *Campylobacter* between 1–24 months of age | Estimate: 0.17 (-0.56, 0.89), p=0.65 |
| **b. ETEC** | | | | | |
| **Weight Gain Outcomes** | | | | | |
| **Change/difference in WAZ** | | | | | **Mean △ in WAZ(95% CI)** |
| | 1 month | Diaz 2023 [19] | N/A | Change in WAZ over 1 month with presence of any ETEC, ST or ST/LT ETEC, and LT ETEC | ETEC: -0.04 (SE: 0.12), p=0.736; ST or ST/LT ETEC: 0.15 (SE: 0.19), p=0.426; LT ETEC: -0.15 (SE: 0.15), p=0.299 |

*(Continued)*

 

| Outcome | Exposure/out-come interval | Paper | Study (if applicable) | Comparison groups | Effect measure |
|---|---|---|---|---|---|
| | | Pajuelo 2024 [33] | | Change in WAZ in the following 30 day interval, comparing the occurrence of an ETEC diarrhea episode to no infection | Coefficient: 0.061 (0.015, 0.108), p=0.009 |
| | 2 months | Das 2022 [17] | GEMS | Within child comparison between baseline and 60 days, by heat stable (est) and heat-labile (elt) serotypes | est: −0.24 (−0.30, −0.18), p<0.001; elt: −0.04 (−0.11, 0.03) p=0.091 |
| | 5 months | Platts-Mills 2017 [36] | PROVIDE | Comparing children with and without LT-ETEC detected | Not reported (see Fig 3#); P-value reported to be not significant (but exact value not reported) |
| | 6 months | George 2023 [22] | REDUCE | Comparing children with and without ETEC detected, by 1.) presence vs. absence OR 2.) log-transformed presence vs. absence | 1.) 0.04 (−0.27 to 0.34) 2.) 0.003 (−0.01 to 0.02) |
| | 12 months | Kabir 2022 [28] | SEEM | Change in WAZ at 12 months among those 1.) with versus 2.) without ETEC detected at 3–6 months | 1.) −0.55 (IQR: −1.40, 0.46) vs. 2.) −0.45 (IQR: −1.14, 0.43), p=0.560 |
| | | [28] | | Change in WAZ at 12 months among those 1.) with versus 2.) without ETEC detected at 9 months | 1.) −0.34 (IQR: −1.27, 0.57) vs. 2.) −0.52 (IQR: −1.20, 0.39), p=0.500 |
| | 24 months | Caulfield 2017 [15] | MAL-ED | Comparing children with LT-ETEC detected in every period compared to children with no LT-ETEC exposure (with periods defined as enrollment to 2 months, 3–5 months, 6–8, 9-11m, 12-17m, and 18-24m) | Not reported but narrative states "tending towards lower growth velocities, no consistent, long-term relationship between LT-ETEC and growth were found" |
| | | Palit 2022 [34] | MAL-ED | Per each ETEC infection detected during routine monthly stool collection | -0.65 (-0.78, -0.42), p=0.02 |
| | | Rogawski 2018 [37] | MAL-ED | comparing high (90th percentile) and low (10th percentile) ETEC prevalence in non-diarrheal stools | Not reported (see Figure S7#); confident interval crosses zero (p-value not significant) |
| | | Kabir 2022 [28] | SEEM | Change in WAZ at 24 months among those 1.) with versus 2.) without ETEC detected at 3–6 months | 1.) −0.50 (IQR: −1.11, 0.46) vs. 2.) −0.42 (IQR: −1.06, 0.51), p=0.850 |
| | | Kabir [28] | | Change in WAZ at 24 months among those 1.) with versus 2.) without ETEC detected at 9 months | 1.) −0.32 (IQR: −1.04, 0.61)vs. 2.) −0.44 (IQR: −1.11, 0.41), p=0.480 |
| | | Pajuelo 2024 [33] | | Change in WAZ at 24 months per change in total number of ETEC episodes over follow-up | Coefficient=0.032 (−0.074, 0.139), p=0.547 |
| **Mean Change in Weight (grams)** | | | | | |
| | 2 months | Lee 2014 [30] | | Per incident episodes of ETEC diarrhea | -25.4 (95%CI: -69.1, 18.3); p=0.255 |
| | | | | Per % of days spent with ETEC diarrhea | -4.5 (95%CI: -9.8, 1.0); p=0.098 |
| | | Black 1984 [13] | | By % of days with ETEC diarrhea | Beta: -0.003 kg, p<0.05 |
| **Risk of Underweight (WAZ<-2)** | | | | | **Odds Ratio (95% CI)** |
| | 9 months | George 2018 [21] | | Comparing likelihood of stunting during follow-up among those with ETEC at baseline to those without, after adjusting for age, age, caregiver educational level, breastfeeding, and family size | 1.27 (0.60, 2.67); p>=0.05 |

*(Continued)*

| Outcome | Exposure/out-come interval | Paper | Study (if applicable) | Comparison groups | Effect measure |
|---|---|---|---|---|---|
| | 24 months | Haque 2023 [25] | MAL-ED | Comparing likelihood of being underweight during follow-up among those with versus without diagnosed ST-ETEC at the following sites: 1.) Bangladesh, 2.) India, 3.) Nepal, 4.) Peru, 5.) Pakistan 6.) South Africa, 7.) Tanzania | 1. 1.10 (0.98, 1.24), p=0.120<br>2. 1.11 (0.98, 1.26), p=0.110<br>3. 1.10 (0.87, 1.40), p=0.415<br>4. 1.07 (0.82, 1.39), p=0.620<br>5. 1.14 (0.98, 1.34), p=0.099<br>6. 1.19 (0.83, 1.72), p=0.349<br>7. 1.27 (1.08, 1.48), p=0.003 |
| | | [25] | | Comparing likelihood of being underweight during follow-up among those with versus without diagnosed LT-ETEC at the following sites: 1.) Bangladesh, 2.) India, 3.) Nepal, 4.) Peru, 5.) Pakistan 6.) South Africa, 7.) Tanzania | 1. 1.10 (0.95, 1.26), p=0.204<br>2. 1.06 (0.95, 1.19), p=0.297<br>3. 0.93 (0.73, 1.20), p=0.593<br>4. 1.02 (0.82, 1.27), p=0.860<br>5. 1.03 (0.88, 1.21), p=0.734<br>6. 0.84 (0.63, 1.13), p=0.258<br>7. 1.07 (0.91, 1.26), p=0.424 |
| | | [25] | | Comparing likelihood of being underweight during follow-up among those with versus without diagnosed ETEC at the following sites: 1.) Bangladesh, 2.) India, 3.) Nepal, 4.) Peru, 5.) Pakistan 6.) South Africa, 7.) Tanzania | 1. 1.15 (1.03, 1.29), p=0.014<br>2. 1.10 (1.00, 1.21), p=0.042<br>3. 1.02 (0.84, 1.22), p=0.872<br>4. 1.05 (0.87, 1.27), p=0.618<br>5. 1.09 (0.97, 1.23), p=0.143<br>6. 0.96 (0.76, 1.22), p=0.747<br>7. 1.33 (1.14, 1.55), p=0.000 |
| | 54–66 months | González-Fernández 2023 [23] | MAL-ED | Univariate analysis comparing relative risk of underweight at follow-up between those with versus without ETEC at least once between 0–5 months of age, adjusted for gender, first weight and income | Relative Risk: 0.72 (0.49, 1.06), p=0.098 |
| **Wasting Outcomes** | | | | | |
| **Mean Change/difference in WHZ** | | | | | **Mean △ in WHZ (95% CI)** |
| | 1 month | Diaz 2023 [19] | N/A | Change in WHZ over 1 month with presence of any ETEC, ST or ST/LT ETEC, and LT ETEC | ETEC: 0.001 (SE: 0.18), p=0.993; ST or ST/LT ETEC: 0.23 (SE: 0.28), p=0.402; LT ETEC: -0.14 (SE: 0.22), p=0.517 |
| | | Pajuelo 2024 [33] | | Change in WHZ in the following 30 day interval, comparing the occurrence of an ETEC diarrhea episode to no infection | Coefficient: 0.060 (0.007, 0.114), p=0.027 |
| | 2 months | Das 2022 [17] | GEMS | Within child comparison between baseline and 60 days, by heat stable (est) and heat-labile (elt) serotypes | est: −0.29 (−0.38, −0.21), p<0.001; elt: −0.05 (−0.11, 0.03), p=0.111 |
| | 6 months | George 2023 [22] | REDUCE | Comparing children with and without ETEC detected, by 1.) presence vs. absence OR 2.) log-transformed presence vs. absence | 1.) −0.26 (−0.60 to 0.09)<br>2.) 0.0004 (−0.02 to 0.02) |
| | 24 months | Rogawski 2018 [37] | MAL-ED | Comparing high (90th percentile) and low (10th percentile) ETEC in non-diarrheal samples | Not reported (See Figure S7#), confident interval crosses zero (p-value not significant) |
| **Risk of Wasting (WHZ<-2)** | | | | | **Odds Ratio (95% CI)** |
| | 9 months | George 2018 [21] | | Comparing likelihood of stunting during follow-up among those with ETEC at baseline to those without | 0.73 (0.19, 2.86); p>=0.05 |
| | 24 months | Haque 2023 [25] | MAL-ED | Comparing likelihood of wasting during follow-up among those with versus without diagnosed ST-ETEC at the following sites: 1.) Bangladesh, 2.) India, 3.) Nepal, 4.) Peru, 5.) South Africa, 6.) Tanzania | 1. 1.07 (0.87, 1.33), p=0.513<br>2. 1.05 (0.86, 1.28), p=0.621<br>3. 1.43 (0.91, 2.24), p=0.118<br>4. 0.90 (0.45, 1.83), p=0.780<br>5. 1.80 (0.93, 3.48), p=0.080<br>6. 2.36 (1.33, 4.19), p=0.003 |

*(Continued)*

| Outcome | Exposure/out-come interval | Paper | Study (if applicable) | Comparison groups | Effect measure |
|---|---|---|---|---|---|
| | | [25] | | Comparing likelihood of wasting during follow-up among those with versus without diagnosed LT-ETEC at the following sites: 1.) Bangladesh, 2.) India, 3.) Nepal, 4.) Peru, 5.) South Africa, 6.) Tanzania | 1. 0.92 (0.70, 1.20), p=0.538<br>2. 1.20 (1.01, 1.42), p=0.034<br>3. 0.92 (0.56, 1.53), p=0.758<br>4. 1.14 (0.66, 1.97), p=0.641<br>5. 1.32 (0.76, 2.27), p=0.322<br>6. 0.82 (0.43, 1.58), p=0.561 |
| | | [25] | | Comparing likelihood of wasting during follow-up among those with versus without diagnosed ETEC at the following sites: 1.) Bangladesh, 2.) India, 3.) Nepal, 4.) Peru, 5.) South Africa, 6.) Tanzania | 1. 1.02 (0.83, 1.24), p=0.857<br>2. 1.17 (1.01, 1.35), p=0.035<br>3. 1.18 (0.82, 1.71), p=0.367<br>4. 1.04 (0.65, 1.69), p=0.860<br>5. 1.53 (0.98, 2.39), p=0.061<br>6. 2.14 (1.12, 4.09), p=0.021 |
| **Linear Growth Outcomes** | | | | | |
| **Mean Change/difference in HAZ/LAZ** | | | | | **Mean △ in HAZ/LAZ (95% CI)** |
| | 1 month | Diaz 2023 [19] | N/A | Change in HAZ over 1 month with presence of any ETEC, ST or ST/LT ETEC, and LT ETEC | ETEC: -0.15 (SE: 0.12), p=0.193; ST or ST/LT ETEC: 0.03 (SE: 0.18), p=0.860; LT ETEC: -0.26 (SE: 0.15), p=0.076 |
| | 2 months | Nasrin 2021 [32] | GEMS | Comparing children with MSD-attributed to ST-ETEC to those with non ST-ETEC-attributed MSD | 0-11 months: 0.006 (95%CI: -0.07, 0.08), p=0.890<br>12-23 months: -0.12 (95%CI: -0.17, -0.06), p<0.001 |
| | | Das 2022 [17] | GEMS | Within child comparison between baseline and 60 days, by heat stable (est) and heat-labile (elt) serotypes | est: −0.09 (−0.14, −0.03), p=0.004; elt: −0.04 (−0.10, 0.04), p=0.161 |
| | 3 months | Platts-Mills 2014 [35] | MAL-ED | Per log10 higher quantity of ETEC infection in asymptomatic stool | -0.15; p-value<0.001 |
| | | Rogawski 2018 [37] | MAL-ED | Within child comparison between baseline and 3 months following ETEC-attributable diarrhea | -0.04 (-0.07, -0.01), NR |
| | 6 months | George 2023 [22] | REDUCE | Comparing children with and without ETEC detected, by 1.) presence vs. absence OR 2.) log-transformed presence vs. absence | 1.) 0.11 (−0.19 to 0.42)<br>2.) −0.01 (−0.03 to 0.01) |
| | 12 months | Donowitz 2021 [20] | | Per 1 episode of diarrhea attributed to ST-ETEC | 0.09 (-0.18, 0.36); p=0.51 |
| | | Schnee 2018 [41] | PROVIDE | Per ST-ETEC -attributable diarrhea episode | 0.02 (95% CI: −0.17, 0.20); p-value NR |
| | | Kabir 2022 [28] | SEEM | Change in HAZ at 12 months among those 1.) with versus 2.) without ETEC detected at 3–6 months | 1.) −0.67 (IQR: −1.41, 0.07) vs. 2.) −0.53 (IQR: −1.48, 0.15), p=1.000 |
| | | [28] | | Change in HAZ at 12 months among those 1.) with versus 2.) without ETEC detected at 9 months | 1.) −0.70 (IQR: −1.61, 0.18) vs. 2.) −0.50 (IQR: −1.31, −0.04), p=0.520 |
| | 18 months | Iqbal 2019 [27] | | Comparing children with and without LT-ETEC at 6 months of age | -0.03; SE: 0.16; p-value=0.842 |
| | | | | Comparing children with and without LT-ETEC at 9 months of age | 0.23; SE: 0.16; p-value 0.146 |
| | | | | Comparing children with and without STh-ETEC at 6 months of age | -0.54; SE 01.6, p-value 0.107 |
| | | | | Comparing children with and without STh-ETEC at 9 months of age | 0.06; SE: 0.27, p-value 0.82 |
| | | | | Comparing children with and without STp-ETEC at 6 months of age | -0.05; SE 0.27, p=0.846 |

*(Continued)*

| Outcome | Exposure/out-come interval | Paper | Study (if applicable) | Comparison groups | Effect measure |
|---|---|---|---|---|---|
| | | | | Comparing children with and without STp-ETEC at 9 months of age | -0.37; SE 0.27, p = 0.158 |
| | 24 months | Caulfield 2017 [15] | MAL-ED | Comparing children with LT-ETEC detected in every period compared to children with no LT-ETEC exposure (with periods defined as enrollment to 2 months, 3–5 months, 6–8, 9-11m, 12-17m, and 18-24m) | NR (narrative states no consistent relationship) |
| | | Donowitz 2021 [20] | | Per 1 episode of diarrhea attributed to ST-ETEC | 0.12 (-0.12, 0.36); p = 0.33 |
| | | Palit 2022 [34] | MAL-ED | Per each ETEC infection detected during routine monthly collection | -0.21 (95%CI: -0.24, 0.10), p < 0.001 |
| | | Rogawski 2018 [37] | MAL-ED | Comparing high (90th percentile) and low (10th percentile) ETEC prevalence in non-diarrheal samples | 0.10 (95%CI: -0.11, 0.31) |
| | | | | Comparing high (90th percentile) and low (10th percentile) ETEC-attributable diarrhea burden | Not reported (see Figure S4#), confident interval crosses zero (p-value not significant) |
| | | | | Per one log increase in ETEC quantity in non-diarrhea stool | 0.02 (-0.03, 0.07) |
| | | Schnee 2018 [41] | PROVIDE | per ST-ETEC attributable episode | 0.01 (95% CI: -0.19, 0.22); p-value NR |
| | | Kabir 2022 [28] | SEEM | Change in HAZ at 24 months among those 1.) with versus 2.) without ETEC detected at 3–6 months | 1.) −0.78 (IQR: −1.46, 0.01)vs. 2.) −0.63 (IQR: −1.48, 0.04), p = 0.970 |
| | | [28] | | Change in HAZ at 24 months among those 1.) with versus 2.) without ETEC detected at 9 months | 1.) −0.63 (IQR: −1.53, 0.20) vs. 2.) −0.74 (IQR: −1.46, 0.00), p = 0.850 |
| | | Pajuelo 2024 [33] | | Change in LAZ at 24 months per change in total number of ETEC episodes over follow-up | Coefficient = 0.045 (−0.053, 0.144), p = 0.366 |
| | 60 months | Rogawski 2018 [37] | MAL-ED | Comparing high (90th percentile) and low (10th percentile) ETEC prevalence in non-diarrheal samples | Not reported (see Fig 4A#), confident interval crosses zero (p-value not significant) |
| | | | | Per one log increase in the mean quantity of ETEC per gram of stool | Not reported (See Fig 4B#), confident interval crosses zero (p-value not significant) |
| **Mean Change in Linear Growth (cm)** | | | | | |
| | 9 months | Lee 2014 [30] | | Per % of days spent with ETEC diarrhea | -0.057 (95%CI: -0.112, -0.002), p = 0.041 |
| | | | | Per incident episode of ETEC diarrhea | -0.029 (95%CI: -0.060, -0.002) |
| **Risk of Stunting (HAZ < -2)** | | | | | **Odds Ratio (95% CI)** |
| | 9 months | George 2018 [21] | | comparing likelihood of stunting during follow-up among those with ETEC at baseline to those without | 1.67 (0.84, 3.32), p>=0.05 |
| | 24 months | Haque 2023 [25] | MAL-ED | Comparing likelihood of stunting during follow-up among those with versus without diagnosed ST-ETEC at the following sites: 1.) Bangladesh, 2.) Brazil, 3.) India, 4.) Nepal, 5.) Peru, 6.) South Africa, 7.) Tanzania | 1. 1.24 (1.11, 1.38), p = 0.000 2. 1.92 (0.69, 5.32), p = 0.209 3. 1.21 (1.06, 1.38), p = 0.006 4. 0.91 (0.74, 1.13), p = 0.401 5. 1.05 (0.90, 1.22), p = 0.570 6. 1.05 (0.82, 1.35), p = 0.697 7. 1.43 (1.28, 1.61), p = 0.000 |

*(Continued)*

| Outcome | Exposure/out-come interval | Paper | Study (if applicable) | Comparison groups | Effect measure |
|---|---|---|---|---|---|
| | | [25] | | Comparing likelihood of stunting during follow-up among those with versus without diagnosed LT-ETEC at the following sites: 1.) Bangladesh, 2.) Brazil, 3.) India, 4.) Nepal, 5.) Peru, 6.) South Africa, 7.) Tanzania | 1. 1.06 (0.93, 1.22), p=0.377<br>2. 0.31 (0.08, 1.31), p=0.112<br>3. 1.14 (1.01, 1.28), p=0.029<br>4. 1.03 (0.84, 1.26), p=0.772<br>5. 0.98 (0.86, 1.11), p=0.707<br>6. 1.22 (1.02, 1.46), p=0.034<br>7. 0.98 (0.87, 1.10), p=0.686 |
| | | [25] | | Comparing likelihood of stunting during follow-up among those with versus without diagnosed ETEC at the following sites: 1.) Bangladesh, 2.) Brazil, 3.) India, 4.) Nepal, 5.) Peru, 6.) South Africa, 7.) Tanzania | 1. 1.25 (1.12, 1.38), p=0.000<br>2. 0.68 (0.29, 1.61), p=0.382<br>3. 1.21 (1.10, 1.34), p=0.000<br>4. 0.97 (0.83, 1.13), p=0.693<br>5. 1.00 (0.90, 1.12), p=0.939<br>6. 1.17 (1.00, 1.36), p=0.046<br>7. 1.33 (1.20, 1.48), p=0.000 |
| **Neurodevelopmental Outcomes** | | | | | |
| **Change in Bayley-III Score** | | | | | |
| | 12 months | Donowitz 2021 [20] | | Per 1 episode of diarrhea attributed to ST-ETEC | Cognition: -0.13 (-1.67 to 1.41), p=0.87<br>Language: 0.49 (-1.36 to 2.35), p=0.6<br>Motor abilities:<br>-0.90 (-2.61 to 0.81), p=0.31 |
| **Change in Semantic Fluency** | | | | | |
| | 72-96 months | Scharf 2023 [40] | MAL-ED | Comparison of semantic fluency (Words/animals in a minute from NEPSY) between children with 1 or more episodes of diarrhea attributable to ETEC and children with 0 episodes attributable to ETEC between 1–24 months of age | Estimate: -0.26 (-0.70, 0.11), p=0.16 |
| **Change in Phonemic Fluency** | | | | | |
| | 72-96 months | Scharf 2023 [40] | MAL-ED | Comparison of phonemic fluency (Words beginning with S & F in a minute from NEPSY) between children with 1 or more episodes of diarrhea attributable to ETEC and children with 0 episodes attributable to ETEC between 1–24 months of age | Estimate: 0.09 (-0.30, 0.48), p=0.64 |
| **Change in Reasoning Skills** | | | | | |
| | 72-96 months | Scharf 2023 [40] | MAL-ED | Comparison of reasoning skills (Raven colored progressive matrices) between children with 1 or more episodes of diarrhea attributable to ETEC and children with 0 episodes attributable to ETEC between 1–24 months of age | Estimate: 0.05 (-0.46, 0.35), p=0.80 |
| **c. Norovirus** | | | | | |
| **Weight Gain Outcomes** | | | | | |
| **Change/difference in WAZ** | | | | | **Mean △ in WAZ(95% CI)** |
| | 2 months | Das 2024 [18] | GEMS | Within child comparison between baseline and 60 days among symptomatic MSD children and asymptomatic children | Symp: 0.15 (0.08, 0.22), p<0.001; Asymp: −0.12 (−0.18,−0.06), p<0.001 |
| | 5 months | Platts-Mills 2017 [36] | PROVIDE | Comparing children with and without norovirus detected | Not reported (see Fig 3#); P-value reported to be not significant (but exact value not reported); |

*(Continued)*

| Outcome | Exposure/out-come interval | Paper | Study (if applicable) | Comparison groups | Effect measure |
|---|---|---|---|---|---|
| | 24 months | Rogawski 2018 [37] | MAL-ED | Comparing high (90th percentile) and low (10th percentile) norovirus prevalence in non-diarrheal stools | Not reported (see Figure S7#); confident interval crosses zero (p-value not significant) |
| | | Palit 2022 [34] | MAL-ED | Per each Norovirus GI infection detected during routine monthly stool collection | -0.04 (-0.12, 0.35), p = 0.281 |
| | | | | Per each Norovirus GII infection detected during routine monthly stool collection | -0.39 (-0.49, -0.28), p < 0.001 |
| | 54–66 months | González-Fernández 2023 [23] | MAL-ED | Univariate analysis comparing differences in WAZ between those with versus without norovirus at least once between 0–11 months of age, adjusted for gender, first weight and income | -0.19 (-0.45, 0.07), p = 0.152 |
| | | [23] | | Univariate analysis comparing differences in WAZ between those with versus without norovirus at least once between 0–5 months of age, adjusted for gender, first weight and income | -0.19 (-0.41, 0.02), p = 0.078 |
| **Risk of Underweight (WAZ<-2)** | | | | | **Odds Ratio (95% CI)** |
| | 54–66 months | González-Fernández 2023 [23] | MAL-ED | Univariate analysis comparing relative risk of underweight at follow-up between those with versus without norovirus at least once between 0–11 months of age, adjusted for gender, first weight and income | Relative Risk: 1.39 (0.93, 2.08), p = 0.107 |
| **Wasting Outcomes** | | | | | |
| **Mean Change/difference in WHZ** | | | | | **Mean △ in WHZ (95% CI)** |
| | 2 months | Das 2024 [18] | GEMS | Within child comparison between baseline and 60 days among symptomatic MSD children and asymptomatic children | Symp: 0.13 (0.05, 0.21), p=0.002; Asymp: −0.13 (−0.19, −0.06), p < 0.001 |
| | 24 months | Rogawski 2018 [37] | MAL-ED | Comparing high (90th percentile) and low (10th percentile) norovirus in non-diarrheal samples | Not reported (See Figure S7#), confidence interval crosses zero (p-value not significant) |
| | 54–66 months | González-Fernández 2023 [23] | MAL-ED | Univariate analysis comparing differences in WHZ between those with versus without norovirus at least once between 0–11 months of age, adjusted for gender, first weight and income | -0.07 (-0.52, -0.02), p = 0.036 |
| | | [23] | | Multivariate analysis comparing differences in WHZ between those with versus without norovirus at least once between 0–11 months of age, adjusted for gender, first weight, income, feeding practices, among others | -0.19 (−0.44, −0.05), p = 0.127 |
| **Linear Growth Outcomes** | | | | | |
| **Mean Change/difference in HAZ/LAZ** | | | | | **Mean △ in HAZ/LAZ (95% CI)** |
| | 2 months | Das 2024 [18] | GEMS | Within child comparison between baseline and 60 days among symptomatic MSD children and asymptomatic children | Symp: 0.1 (0.03, 0.17), p=0.01; Asymp: −0.05 (−0.11, 0.01), p=0.11 |
| | 3 months | Rogawski 2018 [37] | MAL-ED | within child comparison between baseline and 3 months following norovirus-attributable diarrhea | -0.04 (-0.08, 0.00); p-value NR |
| | 6 months | Luoma 2023 [31] | iLiNS-DYAD-M | Difference in LAZ at 24 months comparing children with positive vs negative test result for norovirus at 18 months | −0.15 (−0.47 to 0.17), p=0.35 (positive: −1.62±1.04 vs. negative: −1.78±1.05) |
| | 12 months | Donowitz 2021 [20] | | Per 1 episode of diarrhea attributed to norovirus | 0.31 (-0.31, 0.75); p=0.08 |
| | | Schnee 2018 [41] | PROVIDE | Per norovirus GII-attributable episode of diarrhea | 0.23 (95% CI: −0.02, 0.49); p-value NR |

*(Continued)*

| Outcome | Exposure/out-come interval | Paper | Study (if applicable) | Comparison groups | Effect measure |
|---|---|---|---|---|---|
| | 18 months | Iqbal 2019 [27] | | Comparing children with and without norovirus GI/GII infection at 6 months of age | -0.22; SE: 0.15; p-value=0.133 |
| | | | | Comparing children with and without norovirus GI/GII infection at 9 months of age | -0.29; SE: 0.15; p-value 0.05 |
| | 24 months | Donowitz 2021 [20] | | Per 1 episode of diarrhea attributed to norovirus GII | 0.42 (0.04, 0.80); p=0.03 |
| | | Palit 2022 [34] | MAL-ED | Per each Norovirus GI infection detected during routine monthly stool collection | -0.53 (-0.73, -0.50), p<0.001 |
| | | | | Per each Norovirus GII infection detected during routine monthly stool collection | -0.18 (-0.23, 0.07), p=0.671 |
| | | Schnee 2018 [41] | PROVIDE | per norovirus GII-attributable diarrhea episode | 0.29 (95% CI: 0.00, 0.57); p-value NR |
| | | Rogawski 2018 [37] | MAL-ED | Comparing high (90th percentile) and low (10th percentile) norovirus prevalence in non-diarrheal samples | -0.06 (95%CI: -0.19, -0.07), p-value NR |
| | | | | Comparing high (90th percentile) and low (10th percentile) norovirus-attributable diarrhea burden | Not reported (see Figure S4#) confident interval crosses zero (p-value not significant) |
| | | | | Per one log increase in the mean quantity of norovirus per gram of stool | -0.05 (-0.12, 0.02),p-value nR |
| | 60 months | Rogawski 2018 [37] | MAL-ED | Comparing high (90th percentile) and low (10th percentile) norovirus prevalence in non-diarrheal samples | Not reported (see Fig 4A#), confident interval crosses zero (p-value not significant) |
| | | | | Per one log increase in the mean quantity of norovirus per gram of stool | Not reported (See Fig 4B#), confidence interval does not appear to not cross 0 (statistically significant) |
| **Risk of Stunting (HAZ<-2)** | | | | | **Odds Ratio (95% CI)** |
| | 2 months | Bray 2019 [14] | | Comparing odds of stunting at 2 months follow-up vs. baseline among children with norovirus (moderate-to-severe diarrhea [MSD] cases and controls) | MSD: 0.64 (0.35, 1.17) Controls: 0.84 (95%CI: 0.52, 1.35) |
| **Neurodevelopmental outcomes** | | | | | |
| **Change in Bayley-III Ocore** | | | | | |
| | 12 months | Donowitz 2021 [20] | | Per 1 episode of diarrhea attributed to norovirus GII | Cognition: 2.46 (0.05 to 4.87), p=0.05 Language: 1.00 (-1.93 to 3.93), p=0.51 Motor abilities: 2.48 (-0.26 to 5.23), p=0.08 |
| **Change in Semantic Fluency** | | | | | |
| | 72-96 months | Scharf 2023 [40] | MAL-ED | Comparison of semantic fluency (Words/animals in a minute from NEPSY) between children with 1 or more episodes of diarrhea attributable to norovirus and children with 0 episodes attributable to norovirus between 1–24 months of age | Estimate: 0.18 (-0.22, 0.62), p=0.35 |
| **Change in Phonemic Fluency** | | | | | |
| | 72-96 months | Scharf 2023 [40] | MAL-ED | Comparison of phonemic fluency (Words beginning with S & F in a minute from NEPSY) between children with 1 or more episodes of diarrhea attributable to norovirus and children with 0 episodes attributable to norovirus between 1–24 months of age | Estimate: -0.04 (-0.45, 0.36), p=0.83 |

*(Continued)*

**Table 2.** (Continued)

| Outcome | Exposure/out-come interval | Paper | Study (if applicable) | Comparison groups | Effect measure |
|---|---|---|---|---|---|
| **Change in Reasoning Skills** | | | | | |
| | 72-96 months | Scharf 2023 [40] | MAL-ED | Comparison of reasoning skills (Raven colored progressive matrices) between children with 1 or more episodes of diarrhea attributable to ETEC and children with 0 episodes attributable to ETEC between 1–24 months of age | Estimate: -0.21 (-0.63, 0.21), p=0.33 |

Abbreviations: HAZ, height-for-age z-score; NS, not specified, SD, standard deviation; SE, standard error.

*Represents the number of children enrolled.

#From the original manuscript.

p<0.001) [34]. The other MAL-ED cohort publication did not find an association between high (90th percentile) prevalence of norovirus in non-diarrheal stools and WAZ [37] nor did the analysis from the PROVIDE study comparing change in WAZ five months after norovirus (or no norovirus) detection [36]. A GEMS secondary analysis found discrepant associations between norovirus detection and WAZ 60-days later- norovirus MSD cases gained WAZ and asymptomatic controls with norovirus infection lost WAZ [18]. The two studies from MAL-ED found no evidence of association between norovirus and ponderal growth [23,37].

Of the 18 associations reported on the relationship of norovirus on LAZ/HAZ/stunting from eight unique publications, four were statistically significant and effect sizes for associations with change in LAZ/HAZ ranged from -0.53 to 0.31. For example, mean change in LAZ between baseline and three months following norovirus-attributed diarrhea was associated with a 0.04 lower delta LAZ (95% CI: -0.08, 0.00) [37]. Another analysis from MAL-ED found an association between each norovirus GI infection detected during routine monthly stool over a 24 month period and change in mean change LAZ (-0.53 [95%CI: -0.73, -0.50]) but not norovirus GII infection (-0.18 [95%CI: -0.23, 0.07]) [34]. Each episode of Norovirus GII-attributed diarrhea was associated with a higher delta LAZ at 24 months (0.04 [95%CI: 0.04, 0.80]) [20]. Of the three pathogens evaluated in the review, norovirus (specifically norovirus GII) was the only pathogen significantly associated with a neurodevelopmental outcome, albeit in a positive direction: each episode of norovirus GII diarrhea was associated with a 2.46 higher Cognitive Bayley score (95%CI: 0.05, 4.87) [20] but had no association with semantic or phonemic fluency or reasoning skills [40].

## Discussion

The consequences of enteric pathogens beyond diarrhea are an important consideration for vaccine development prioritization. In this systematic review of observational studies following children with and without *Campylobacter*, ETEC, and/or norovirus infections, we found modest evidence of linear growth detriments associated with all three pathogens, modest evidence of *Campylobacter* impacting weight, and no evidence of detrimental impacts of these pathogens on wasting or neurodevelopment, albeit these two outcomes were rarely reported. Differences in outcome definitions, comparison groups, and timeframes prohibited meta-analysis and emphasize the need for more standardized reporting of anthropometric and neurocognitive outcomes following enteric pathogen exposure. Because these outcomes have multiple causes and occur over long-time frames, these associations are particularly prone to confounding, reverse causality, and selection bias. Ultimately, highly efficacious randomized controlled trials of interventions targeting specific enteric pathogen infection and disease are needed to establish the magnitude and relative importance of long-term consequences from enteric pathogens.

Studies conducted in the same setting frequently had discrepant results. For example, three studies evaluated linear growth over 12 months associated with *Campylobacter* in Bangladesh [20,39,41]. Some found an association with the

pathogen and outcome in the presence [41] and absence of diarrhea [39], while others did not [20]. We also found several publications utilizing the same underlying dataset (such as from MAL-ED) but resulting in variable conclusions. For example, Palit (2022) [34], Caulfield (2017) [15], and Rogawski [37] (2018) all reported on changes in LAZ 24 months after enrollment associated with ETEC from all country sites in the MAL-ED study, with change in HAZ/LAZ estimates ranging from 0.10 to -0.21 and no statistically significant association [15,37] to a statistically significant association [34]. While these publications captured different pathogen exposure classifications (per each ETEC infection during monthly stool collection; comparing high to low prevalence in monthly stool; children with and without asymptomatic ETEC per 3 month period) and two studies used molecular methods while one used culture-based methods, the large variation in results from the same cohort was surprising. As definitive answers around relative contribution of specific pathogens to long-term outcomes is necessary to prioritize resources, consensus for the most relevant/ interpretable exposure classification and outcomes measures/timepoints would enable more efficient and interpretable evidence synthesis.

Exposure classification (per % day spent with pathogen-specific diarrhea, per incident episode of pathogen-specific diarrhea, with/without pathogen of interest at a single, prior time point, high [90th percentile] to low [10th percentile] pathogen-attributable diarrhea burden, per one log increase in cumulative pathogen quantity, per each pathogen detection over several time points) was not the only source of heterogeneity between, and within, studies. Differences in diagnostics (ELISA, culture, or the most sensitive, qPCR) for pathogen detection; detection of pathogens in diarrhea vs. asymptomatic fecal samples, differences in timeframes (ranging from 2 to 60 months), differences in outcome/dependent variable framing (for example, change in LAZ, single time point LAZ, change in growth (cm), odds of stunting), and differences in confounder adjustment make cross-study comparisons challenging. Individual patient data meta-analyses, an increasingly appreciated gold-standard in evidence synthesis, could overcome some challenges with sources of heterogeneity, such as comparison groups and confounders, as long as comparable information is available between studies.

Overcoming measured and unmeasured confounders remains a challenge in observational studies of long-term processes. Randomizing children to highly efficacious pathogen specific vaccines or treatment and evaluating outcomes (vaccine or treatment probe studies) will enable more robust causal inference in establishing enteric pathogen-attributable long-term morbidities by preventing confounding. The Antibiotics for Children with severe Diarrhea (ABCD) trial, a 7-country double-blind placebo controlled trial testing the efficacy of azithromycin among children with watery diarrhea and other severity indicators, found 3-days of azithromycin treatment to reduce the loss in LAZ 90-days following ST-ETEC attributed diarrhea by 0.08 (95%CI: 0.01, 0.14) compared to placebo among 889 children [42]. Too few children had *Campylobacter* identified in this trial to conduct this same probe. Whether the short-term improvements in LAZ associated with azithromycin are sustained following ST-ETEC infection, and whether such improvements translate to important developmental milestones, like neurocognitive development, remains unknown. Antibiotic probe studies are a useful tool for interrogating bacterial pathogen consequences, but to the best of our knowledge, no such treatment probes exist for viral infections such as norovirus.

While pathogen-specific vaccine trials offer the best way to obtain unconfounded estimates of long-term pathogen-specific effects, they will not be without challenge. Vaccines take decades to be developed. The most advanced vaccine candidate for ETEC is in phase 2 clinical development, phase 3 for norovirus, and to the best of our knowledge, there are no candidates for *Campylobacter* at this time. Phase 3 vaccine trials are cost-prohibitive, particularly those that include lengthy follow-up to accrue long-term outcomes. Vaccines may not cover all serotypes of a given pathogen, may only prevent disease and not infection, and/or may be sub-optimally effective. Furthermore, population-level effect sizes of interventions, such as vaccines, on longer-term outcomes are expected to be small because only a minority of children in the trial will be infected with the targeted pathogen and even fewer will develop disease from said pathogen. Therefore, the overall vaccine effect will be heavily diluted by children who were not at risk for vaccine preventable, pathogen-attributable, long-term outcomes of interest, such as growth or neurodevelopmental faltering. Vaccine trials that have been powered to a primary disease endpoint are likely to be dramatically underpowered for growth and other long-term outcomes.

Longitudinal studies of early childhood diarrhea and longer-term cognitive outcomes have not shown consistent evidence of an association [43]. However, early growth faltering and biomarkers of enteric and systemic inflammation, processes likely causally linked to enteric pathogens, do appear to have stronger ties with cognitive outcomes, albeit variably by site [40]. Because the cost of following a cohort of individually randomized participants for several years is prohibitive, post-vaccine introduction studies and disease surveillance will be required to obtain necessary data to inform long-term impacts of interventions against enteric pathogens. As new enteric pathogen vaccines are being considered for introduction, robust population-level surveys of key outcomes like stunting and neurodevelopment will be critical to have in place so the full value of these vaccines can be informed by real data. The field can draw on other pathogen examples- such as measles which was studied post-vaccine introduction, to further informs the value proposition of this important vaccine [44].

This review had several limitations. To represent the breadth of ways in which enteric pathogens and longer-term outcomes are reported in the literature, we did not restrict studies, nor limit abstracted information, to specific exposure and outcome framings leading to difficulty in interpretation across heterogeneous measures and comparison groups. We additionally allowed for multiple publications from the same underlying cohort to be included in the review when outcomes or comparison groups varied slightly. This approach further added to heterogeneity and difficulty in interpretation. Individual participant data meta-analysis could overcome these challenges, albeit with a significantly larger time and resource investment. Given the challenges with confounding and reverse causality, we do not believe that additional observational studies are needed. Instead, investment in pathogen-targeted randomized controlled intervention trials, and specifically, post-introduction (phase 4) vaccine trials, will likely be the optimal setting for estimation of true pathogen-specific associations with salient long-term of enteric pathogens, such as stunting and neurocognitive delay. Additionally, consensus on the most relevant outcomes to include, such as change in LAZ/HAZ between enrollment and 24 months of follow-up, will hasten future synthesis exercises.

## Supporting information

**S1 Table. Search terms.**
(DOCX)

**S2 Table. Quality assessment scale.**
(DOCX)

**S3 Table. Quality assessment of included studies.**
(DOCX)

**S1 File. PRISMA checklist [14].**
(DOCX)

## Author contributions

**Conceptualization:** Birgitte Giersing, Mateusz Hasso-Agopsowicz.

**Data curation:** Gregory K. Zane, Mathias Lalika, Fatima Al-Shimari, Priyanka Shrestha.

**Formal analysis:** Gregory K. Zane, Mathias Lalika, Fatima Al-Shimari, Priyanka Shrestha.

**Funding acquisition:** Birgitte Giersing.

**Investigation:** Patricia B. Pavlinac, Ibrahim Khalil, Elizabeth T. Rogawski McQuade, James A. Platts-Mills, Mateusz Hasso-Agopsowicz.

**Methodology:** Patricia B. Pavlinac, Gregory K. Zane, Mathias Lalika, Fatima Al-Shimari, Priyanka Shrestha, Mateusz Hasso-Agopsowicz.

**Project administration:** Gregory K. Zane, Mateusz Hasso-Agopsowicz.

**Software:** Gregory K. Zane.

**Supervision:** Patricia B. Pavlinac, Mateusz Hasso-Agopsowicz.

**Validation:** Gregory K. Zane.

**Visualization:** Gregory K. Zane, Mathias Lalika, Priyanka Shrestha, Mateusz Hasso-Agopsowicz.

**Writing – original draft:** Patricia B. Pavlinac, Gregory K. Zane.

**Writing – review & editing:** Gregory K. Zane, Ibrahim Khalil, Elizabeth T. Rogawski McQuade, James A. Platts-Mills, Mathias Lalika, Priyanka Shrestha, Birgitte Giersing, Mateusz Hasso-Agopsowicz.

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
