## [Decision Letter · Decision Letter 0]

7 Oct 2025

Dear Dr. Pavlinac,

We are pleased to inform you that your manuscript 'Anthropometric and neurocognitive consequences of Campylobacter, enterotoxigenic Escherichia coli, and norovirus: A systematic review' has been provisionally accepted for publication in PLOS Neglected Tropical Diseases.

Best regards,

Joseph M. Vinetz

Section Editor

Joseph Vinetz

Section Editor

Shaden Kamhawi

co-Editor-in-Chief

Paul Brindley

co-Editor-in-Chief

Reviewer's Responses to Questions

**Key Review Criteria Required for Acceptance?**

**Methods**

-Are the objectives of the study clearly articulated with a clear testable hypothesis stated?

-Is the study design appropriate to address the stated objectives?

-Is the population clearly described and appropriate for the hypothesis being tested?

-Is the sample size sufficient to ensure adequate power to address the hypothesis being tested?

-Were correct statistical analysis used to support conclusions?

-Are there concerns about ethical or regulatory requirements being met?

Reviewer #1: The objectives, study design, the selection of review papers, review and checking procedure are clearly indicated in methods section

Reviewer #2: Methods are clear and allow the authors to answer their research questions, as this is a review article there are no ethical concerns

**Results**

-Does the analysis presented match the analysis plan?

-Are the results clearly and completely presented?

-Are the figures (Tables, Images) of sufficient quality for clarity?

Reviewer #1: The findings are addressed clearly. Please find the attached my attached comments for further recommendation

Reviewer #2: Result from the literature review are adequately described,

**Conclusions**

-Are the conclusions supported by the data presented?

-Are the limitations of analysis clearly described?

-Do the authors discuss how these data can be helpful to advance our understanding of the topic under study?

-Is public health relevance addressed?

Reviewer #1: Conclusions supported the results and relevant public health importance are clearly addressed.

Reviewer #2: Conclusions are appropriate to the literature review and limitations have been included. Public health relevance is not addressed and the paper does not really advance the field of knowledge.

**Editorial and Data Presentation Modifications?**

Reviewer #1: (No Response)

Reviewer #2: (No Response)

**Summary and General Comments**

Reviewer #1: The exposures and outcomes of the review papers have been adequately addressed. The paper has important public health importance for prevention and control.

Reviewer #2: This paper describes a literature review, the authors had specific search criteria to find and included manuscripts, as a result of this the number if papers per bacterium described are small, limiting the present paper in terms of advancing the field of knowledge. Some papers included used the same dataset and so findings from these papers are similar which is introducing bias into this paper and limits the conclusions that can be drawn from it.

PLOS authors have the option to publish the peer review history of their article (what does this mean?). If published, this will include your full peer review and any attached files.

Reviewer #1: **Yes: **M. Jahangir Hossain

Reviewer #2: No

---

## [Editor Report · Acceptance letter]

Dear Dr. Pavlinac,

We are delighted to inform you that your manuscript, "Anthropometric and neurocognitive consequences of Campylobacter, enterotoxigenic Escherichia coli, and norovirus: A systematic review," has been formally accepted for publication in PLOS Neglected Tropical Diseases.

Best regards,

Shaden Kamhawi

co-Editor-in-Chief

Paul Brindley

co-Editor-in-Chief
